

# Effects of defoliation and nitrogen on carbon dioxide ($CO_2$) emissions and microbial communities in soils of cherry tree orchards

Jing Wang[1], Yibo Wang[1], Ruifang Xue[2], Dandan Wang[2] and Wenhui Nan[2]

[1] Tianshui Normal University, Gansu Key Laboratory of Utilization of Agricultural Solid Waste Resources, Tianshui, China
[2] Tianshui Normal University, College of Bioengineering and Biotechnology, Tianshui, China

## ABSTRACT

**Background**. In farmland, microbes in soils are affected by exogenous carbon, nitrogen, and soil depth and are responsible for soil organic carbon (SOC) mineralization. The cherry industry has been evolving rapidly in northwest China and emerged as a new source of income for local farmers to overcome poverty. Accordingly, it is highly imperative to probe the effect of defoliation and nitrogen addition on carbon dioxide ($CO_2$) emissions and microbial communities in soils of dryland cherry orchards.

**Methods**. $CO_2$ emissions and microbial communities were determined in soil samples at three depths, including 0–10 cm, 10–30 cm, and 30–60 cm, from a 15-year-old rain-fed cherry orchard. The samples were respectively incubated with or without 1% defoliation under three input levels of nitrogen (0 mg kg$^{-1}$, 90 mg kg$^{-1}$, and 135 mg kg$^{-1}$) at 25°C in the dark for 80 days.

**Results**. Defoliation and nitrogen addition affected $CO_2$ emissions and microbial communities and increased microbial biomass carbon (MBC), the activity of soil catalase, alkaline phosphatase, and cellulase in soils of the dryland cherry orchard. The culture with defoliation significantly promoted $CO_2$ emissions in soils at the three depths mainly by increasing the MBC, catalase, alkaline phosphatase, and cellulase activities, resulted in positive priming index. Nitrogen addition elevated the MBC and changed soil enzymes and reduced $CO_2$ emissions in soils at the three depths. Moreover, the priming index was higher in deep soils than in top and middle soils under the condition of defoliation and nitrogen addition. No significant differences were observed in the soil bacterial diversity (Chao1, Shannon, and Simpson) among all treatments. Meanwhile, the relative abundance of *Proteobacteria* was markedly increased and that of *Acidobacteria* was substantially diminished in soils at the three depths by defoliation and nitrogen addition. The results sustained that defoliation and nitrogen can regulate SOC dynamics by directly and indirectly affecting soil microbial activities and communities. As a result, the combination of defoliation return and nitrogen fertilization management is a promising strategy to increase SOC and promote soil quality in dryland cherry orchards.

Corresponding author
Jing Wang, 690958228@qq.com

## INTRODUCTION

Soil respiration is the primary mode of carbon dioxide ($CO_2$) exchange between the soil carbon pool and atmosphere and one of the largest fluxes in the carbon cycle of terrestrial ecosystems (78–98 Gt C per year) (*Bond-Lamberty et al., 2018*; *Tang et al., 2020*). Accordingly, soil respiration can alter $CO_2$ in the atmosphere and carbon storage in soils, thus affecting the global carbon cycle (*Wang et al., 2019*). As a vital component of land respiration in agricultural and forestry ecosystems, soil respiration is readily influenced by vegetation types, cultivation, fertilization, and other active jamming factors (*Buchmann, 2000*; *Raich & Tufekcioglu, 2000*; *Mo et al., 2021*). Reportedly, exogenous carbon inputs affect soil nutrient availability, increase microbial activities and quantity, and change the activity of soil enzymes, thus contributing to the priming effect of $CO_2$ emissions (*Blagodatskaya & Kuzyakov, 2008*; *Chen et al., 2015*; *Li et al., 2018*; *Liang et al., 2022*). Most of the existing studies focused on the association of soil carbon dynamics and microbial communities with labile carbon inputs (*Blagodatskaya et al., 2007*; *Tian et al., 2016a*; *Li et al., 2017*; *Shahbaz et al., 2018*; *Zhou et al., 2021*). In addition, the effects of complex polymerized organism inputs (such as straw and litter) on soil carbon dynamics and microbial communities have been analyzed mainly in farmland and forestry ecosystems (*Moorhead, Sinsabaugh & Moorhead, 2006*; *Aye et al., 2018*; *Fang et al., 2018b*; *Mo et al., 2021*) but rarely in orchard ecosystems (*Zhou et al., 2021*). Therefore, the study of defoliation inputs to orchard soils can objectively and truly reflect the characteristics of soil $CO_2$ emissions and microbes in the orchard industry.

Nitrogen fertilizer plays a pivotal role in the management of orchards, which can increase soil nutrients and change microbial activities and communities (*Neff et al., 2002*; *Malhi et al., 2003*; *Leff et al., 2015*; *Wang, Liu & Bai, 2018*), and affect soil respiration (*Razanamalala et al., 2018*; *Sawada, Inagaki & Toyota, 2021*; *Na et al., 2022*). Of note, some hypotheses have been proposed on the mechanism of nitrogen. For instance, the theory of "Microbial stoichiometric decomposition" (*Hessen et al., 2004*; *Chen et al., 2014*; *Cui et al., 2020*) demonstrates that exogenous substance inputs can contribute to high microbial activities and organic matter decomposition, which simulate high $CO_2$ emissions, by meeting the carbon/nitrogen demand of soil microbes, indicating that high nitrogen availability (abundant nutrients) may favor soil organic matter (SOM) decomposition. The "microbial nitrogen mining" theory shows that soil microbes decompose SOM with labile carbon as an energy source to obtain the required nitrogen and then induce carbon priming effects under low nitrogen conditions (*Moorhead, Sinsabaugh & Moorhead, 2006*; *Borrajo et al., 2011*; *Mason-Jones, Schmücker & Kuzyakov, 2018*; *Na et al., 2022*; *Craine, Morrow & Fierer, 2007*), illustrating that low nitrogen availability (poor nutrients) stimulates SOM decomposition. Despite the obvious odds between the two theories, their mechanisms are closely related to organic carbon, which suggests an inherently critical association of soil carbon respiration with carbon and nitrogen inputs. In this context, there is a need to explore the impacts of exogenous defoliation inputs on $CO_2$ emissions and microbial mechanisms under different nitrogen levels, thus providing more data and theoretical support for understanding the soil carbon cycle.

Responses of soil $CO_2$ emissions to different environments are variable because of different soil environments and physicochemical characteristics in soils at varying depths (*De Graaff et al., 2014*; *Tian et al., 2016b*; *Liao et al., 2020*). In top soils ($\leq$10 cm), large amounts of SOC are produced and $CO_2$ emissions are increased because of litter, fertilization, soil fungi, bacteria, and animals (*Blanco-Canqui & Lal, 2008*; *Tian et al., 2016b*; *Banfield et al., 2018*). Conversely, deep soils can sequestrate more exogenous carbon than top soils because of low SOC (*Fontaine et al., 2007*; *Wang et al., 2014*). Consequently, it is urgent to investigate whether soil respiration in soils at different depths is affected by defoliation and nitrogen additions, thereby providing data to support the development of soil carbon sequestration science.

The cherry industry is highly economically profitable, which has contributed to its rapid growth in northwest China. In this context, this industry has become a new source of income for local farmers to shake off poverty. Accordingly, it is necessary to explore microbial mechanisms of $CO_2$ emissions in dryland cherry orchards with defoliation and nitrogen addition, thus providing more data and theoretical support for sustainable development of cherry industry.

In this study, soils at different depths were collected from a rain-fed cherry orchard in northwest China for an indoor incubation experiment was performed to ascertain the microbial mechanism regulating $CO_2$ emissions under defoliation and nitrogen addition. Then, the following three hypotheses were proposed: (1) defoliation and nitrogen addition promoted $CO_2$ emissions; (2) $CO_2$ emissions were strongly associated with microbial activities and community; (3) microbial communities varied in soils at different depths.

## MATERIAL AND METHODS

### Soil collection

Soil samples were obtained at three depths (0–10 cm [top soils], 10–30 cm [middle soils], and 30–60 cm [deep soils]) of the botanical garden test site at Tianshui Normal University (Tianshui, Gansu, China; 34°34′10″N and 105°41′47″E) in 2021. The test site was built in 2002, where the cherry rootstock was Gisela 5 and the cherry variety was Provence. The basic physical and chemical properties of top, middle, and deep soils were as follows: pH: 6.7, 7.8, and 8.1; total organic carbon: 14.74 g kg$^{-1}$, 12.54 g kg$^{-1}$, and 7.92 g kg$^{-1}$; total nitrogen: 0.74 g kg$^{-1}$, 0.69 g kg$^{-1}$, and 0.46 g kg$^{-1}$; available phosphorus: 4.93 mg kg$^{-1}$, 5.08 mg kg$^{-1}$, and 3.07 mg kg$^{-1}$; available potassium: 153.0 mg kg$^{-1}$, 148.0 mg kg$^{-1}$, and 100.6 mg kg$^{-1}$.

### Experimental design

The incubation experiment was conducted as a complete factorial experiment of soils at three depths (top, middle, and deep soils) * two kinds of defoliation addition (no-defoliation [$C_0$] and defoliation [$C_A$, 1%]) * addition of three levels of nitrogen (0 mg kg$^{-1}$ [$N_0$], 90 mg kg$^{-1}$ [$N_L$], and 135 mg kg$^{-1}$ [$N_H$]) in three replicates with a fully randomized design. Total organic carbon was 451 g kg$^{-1}$ and total nitrogen was 12.47 g kg$^{-1}$ in defoliation.

Incubation tanks (1 L) were respectively added with 100 g soils (dry weight) at different depths for 7 days of pre-incubation at 25 °C (*Mo et al., 2021*). After that, soils were

fully mixed with defoliation according to different treatments. Nitrogen and phosphate fertilizers were dissolved in distilled water and added to soils as a solution. Next, the soils were incubated at 25 °C with the moisture of 60% and the bulk density of 1.2 g cm$^{-3}$ in the dark for 80 days.

## Gas collection and soil sampling

For each treatment, three tanks were taken out at 1, 4, 13, and 80 days. The soils in the incubation tanks were stored at −20 °C for detecting microbial biomass carbon (MBC), soil enzyme activities, and microbial communities. Specifically, MBC was measured with the chloroform-fumigation extraction method (*Vance, Brookes & Jenkinson, 1987*). The activity of alkaline phosphatase was examined with the method of Tabatabai and Bremner (*Tabatabai & Bremner, 1969*) as described in a prior study (*Miralles et al., 2012*). The activity of cellulase was detected using the method of Xu and Zheng (*Xu & Zheng, 1986*). The activity of catalase was determined by titrating 0.1 mol L$^{-1}$ KMnO$_4$ (*Guan, Zhang & Zhang, 1986*).

## DNA extraction and high-throughput sequencing

Total DNA was extracted from soils with the Fast DNA SPIN Kit for Soil and FastPrep-24 nucleic acid Extraction instrument (MP Biomedicals, Santa Ana, CA, USA) on the 4th day. The DNA was examined with 1% agarose gel electrophoresis, and its concentration was measured with NanoDrop 2000. The DNA was stored in a refrigerator at −20 °C for subsequent use. The stock solution of the DNA was diluted to about 5 mg/L as the template of polymerase chain reaction (PCR) amplification. Afterward, the 16S rRNA variable regions V3-V4 of bacteria were subjected to PCR amplification with the following primers: CCTAYGGGRBGCASCAG (341F) and GGACTACNNGGGTATCTAAT (806R). The PCR was conducted with a system of 50 μL, including 5 μL of 10 * buffer, 4 μL of dNTP, 0.5 μL of RTAQ (Takara), 1 μL of 10 μmol L$^{-1}$ each front and rear primers, 36.5 μL of ddH$_2$O, and 2 μL of template DNA, and the detection of each sample was repeated three times. The amplification conditions were as follows: pre-denaturation at 98 °C for 30 s, 30 cycles of chain disassembly at 98 °C for 10 s, annealing at 55 °C for 15 s, and extension at 72 °C for 1 min, and final extension at 72 °C for 10 min. The products obtained by three repeats of DNA amplification were mixed and tested with 1% agarose gel electrophoresis. The concentration of the obtained bacterial PCR products was measured with the PicoGreen kit. After the products were evenly mixed, DNA was purified and recovered with the DNA purification kit (TIANGEN Biotech, Beijing, China). Bacterial 16S rDNA was sequenced by Shenzhen Weicomeng Technology Group Co., Ltd. (Shenzhen, China) with the Novaseq 6000 PE 250 platform.

According to corresponding barcodes, the samples were subjected to paired reading, and then their barcode and primer sequence were removed. The reads at both ends were combined with the FLASH (V1.2.7) software. The QIIME data analysis package was utilized to remove low-quality raw sequences (length < 250 bp, ambiguous base "N", and the average base quality score less than 20). The chimeric sequence was discarded with the MAARJAM database (https://maarjam.ut.ee/). The valid readings were assigned

to operational taxon units or virtual taxa with an identity threshold of 97%, and the representative sequences were identified with SILVA (V128, http://www.arb-silva.de) and MAARJAM databases.

## Calculation

$CO_2$ emissions (mg $CO_2$ $kg_{soil}^{-1}$) were calculated as per the titration results of hydrochloric acids.

$$CO_2 = \frac{(V_0 - V) \times C_{HCl}}{2} \times 12 \times \frac{1}{m(1-a)} \times 1000.$$

In this formula, $V_0$ was the volume of titration by standard hydrochloric acid during blank calibration (mL), while V represented the volume of titration by standard hydrochloric acid during samples (mL). $C_{HCl}$ was the concentration of standard hydrochloric acid (1 mol $L^{-1}$), m was soil mass (g), and a was soil water content (%).

The $CO_2$ efflux rate (mg $CO_2$ $kg_{soil}^{-1}$ $d^{-1}$) was calculated with the following formula: $CO_2$ efflux rate = $CO_2$ emission/t.

In this formula, t represented the day when the NaOH solution was placed in the incubation tanks.

The cumulative $CO_2$ emission (g $CO_2$ $kg_{soil}^{-1}$) was the cumulative $CO_2$ emission from each treatment over a given incubation time.

For a given incubation time (80 days), the priming index (PI) induced by exogenous substance addition was normalized to the proportion of added non-exogenous substances to cumulative $CO_2$ emissions based on the following formula:

Priming index (PI) = ($CO_{2\ add} - CO_{2\ non-add}$)/ $CO_{2\ non-add}$.

In this formula, $CO_{2\ add}$ represented cumulative $CO_2$ emissions after 80 days of defoliation and nitrogen treatment, and $CO_{2\ non-add}$ indicated cumulative $CO_2$ emissions after 80 days of treatment without defoliation and nitrogen.

The PI represented the intensity of the priming effect. The PI value of 1 suggested that the amount of organic carbon mineralization was not affected by exogenous substance addition. The PI value of >1 indicated that the addition of exogenous substances caused the priming effect of organic carbon mineralization, and the larger value was associated with a stronger priming effect. The PI value of <1 represented that the addition of exogenous substances reduced the mineralization of organic carbon and produced a negative priming effect, and the smaller value illustrated the stronger negative priming effect.

## Statistical analysis

All statistical analyses were performed with the SPSS 18.0 statistical and R programming software. The effects of the $CO_2$ efflux rate, cumulative $CO_2$ emission, PI, MBC, catalase, alkaline phosphatase, cellulase, Chao1, Shannon, and Simpson and the relative abundance of *Proteobacteria* and *Acidobacteria* were analyzed with three-way analysis of variance combined with Duncan's multiple range test. Partial least squares discrimination analysis was conducted to analyze the distribution of soil microbial communities in soils at different depths. The Spearman correlation analysis was used to clarify the correlations of defoliation, nitrogen, and soil depth with $CO_2$ emissions, soil microbial activity parameters,
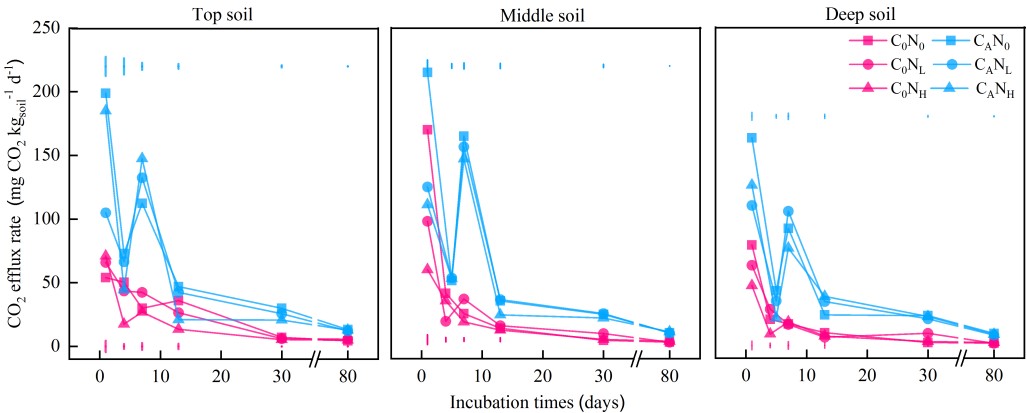

**Figure 1** $CO_2$ efflux rate (mg $CO_2$ kg$_{soil}^{-1}$d$^{-1}$) from the top soil, middle soil, and deep soil in each treatment over the entire incubation period. $C_0$, no-defoliation addition; $C_A$, defoliation addition; $N_0$, no-nitrogen input; $N_L$, low-nitrogen input; $N_H$, high-nitrogen input. The different colored bars show least significant differences (at 5% level) between nitrogen input levels within same defoliation addition at each sampling point.

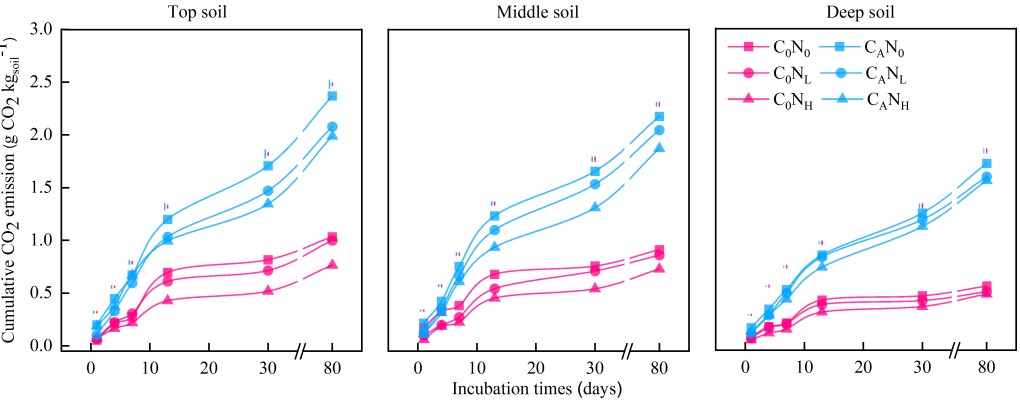

**Figure 2** Cumulative $CO_2$ emission (g $CO_2$ kg$_{soil}^{-1}$) from the top soil, middle soil, and deep soil in each treatment over the entire incubation period. $C_0$, no-defoliation addition; $C_A$, defoliation addition; $N_0$, no-nitrogen input; $N_L$, low-nitrogen input; $N_H$, high-nitrogen input. The different colored bars show least significant differences (at 5% level) between nitrogen input levels within same defoliation addition at each sampling point.

*Proteobacteria* and *Acidobacteria*. Significance was defined at $P < 0.05$. Additionally, Origin18.0 and R programming software were utilized for mapping.

# RESULTS

## CO$_2$ emissions

The $CO_2$ efflux rate and cumulative $CO_2$ emissions were increased under defoliation addition, which was extremely significantly affected by the soil depth, defoliation, and nitrogen, as well as interaction among these three factors (Figs. 1 and 2, and Table 1).

**Table 1 Statistical significance ($P$ value) of the soil depth, defoliation addition, nitrogen input level, and their interaction on the concerned variables.**

| | Soil depth (S) | Defoliation (C) | Nitrogen (N) | S × C | S × N | C × N | S × C × N |
|---|---|---|---|---|---|---|---|
| $CO_2$ efflux rate (mg $CO_2$ $kg_{soil}^{-1}$ $d^{-1}$) | <0.001 | <0.001 | <0.001 | <0.001 | <0.001 | <0.001 | <0.001 |
| Cumulative $CO_2$ emission (g $CO_2$ $kg_{soil}^{-1}$) | <0.001 | <0.001 | <0.001 | <0.001 | <0.001 | <0.001 | <0.001 |
| PI index (80 days) | <0.001 | <0.001 | <0.001 | <0.001 | <0.001 | <0.001 | <0.001 |
| MBC (4th day) | <0.001 | <0.001 | <0.001 | <0.001 | 0.096 | <0.01 | 0.01 |
| Catalase (4th day) | <0.001 | <0.001 | <0.001 | <0.001 | <0.001 | <0.001 | <0.001 |
| Alkaline phosphatase (4th day) | <0.001 | <0.001 | <0.001 | <0.001 | <0.001 | <0.001 | <0.001 |
| Cellulase (4th day) | <0.001 | <0.001 | <0.001 | 0.001 | <0.001 | <0.001 | 0.07 |
| Chao1 index (4th day) | <0.001 | 0.14 | 0.80 | 0.97 | 0.22 | 0.21 | 0.14 |
| Shannon (4th day) | 0.016 | 0.204 | 0.090 | 0.787 | 0.003 | 0.330 | 0.450 |
| Simpson (4th day) | 0.337 | 0.507 | 0.406 | 0.446 | 0.073 | 0.363 | 0.628 |
| Relative abundance of Proteobacteria (%) | <0.001 | <0.001 | <0.001 | <0.001 | 0.001 | 0.05 | 0.003 |
| Relative abundance of Acidobacteria (%) | <0.001 | 0.052 | <0.001 | <0.001 | <0.001 | 0.001 | <0.001 |

The $CO_2$ efflux rate peaked on the 1st day and then declined with the incubation time of each treatment (Fig. 1). Among all treatments, the $CO_2$ efflux rate was higher in $C_A$ treatment (65.41 mg $CO_2$ $kg_{soil}^{-1}$ $d^{-1}$) than in $C_0$ treatment (26.42 mg $CO_2$ $kg_{soil}^{-1}$ $d^{-1}$) and lower in deep soils (36.1 mg $CO_2$ $kg_{soil}^{-1}$ $d^{-1}$) than middle (51.56 mg $CO_2$ $kg_{soil}^{-1}$ $d^{-1}$) and top (50.06 mg $CO_2$ $kg_{soil}^{-1}$ $d^{-1}$) soils (Fig. 1). In addition, $N_0$ treatment (52.87 mg $CO_2$ $kg_{soil}^{-1}$ $d^{-1}$) resulted in a higher $CO_2$ efflux rate than $N_L$ treatment (51.56 mg $CO_2$ $kg_{soil}^{-1}$ $d^{-1}$) and $N_H$ (50.06 mg $CO_2$ $kg_{soil}^{-1}$ $d^{-1}$) on the 1st day.

After 80 days (Fig. 2), cumulative $CO_2$ emissions were averagely elevated by 2.62, 2.37, or 2.26 times under treatments of $C_A N_0$, $C_A N_L$, or $C_A N_H$ when compared to under $C_0 N_0$ treatment. Cumulative $CO_2$ emissions were 30.28% and 28.21% higher in top and middle soils than in deep soils, respectively. Meanwhile, cumulative $CO_2$ emissions were markedly lower under $N_L$ and $N_H$ treatment than under $N_0$ treatment by 7.87% and 15.75%, respectively.

## PI

After 80 days of incubation, defoliation addition alone or in combination with nitrogen addition resulted in positive PI in soils at the three depths (Fig. 3). PI was extremely substantially affected by defoliation, nitrogen, and soil depth (Table 1). PI was the highest in deep soils (2.05 [$C_A N_0$], 1.83 [$C_A N_L$], and 1.76 [$C_A N_H$]) in defoliation and nitrogen additions. Moreover, nitrogen addition alone caused reverse PI in soils at three depths.

## MBC

MBC was enhanced under defoliation and nitrogen addition, which was significantly or extremely significantly altered by defoliation, nitrogen and soil depth (Fig. 4 and Table 1). Moreover, defoliation addition alone or combined with nitrogen addition contributed to higher MBC than $C_0 N_0$ treatment. On the 4th day, MBC in top soils was markedly higher under $C_A N_0$, $C_A N_L$, and $C_A N_H$ treatment than under $C_0 N_0$ treatment by 66.49%, 87.82%, and 97.81%. In contrast, nitrogen addition alone diminished MBC in soils at the three

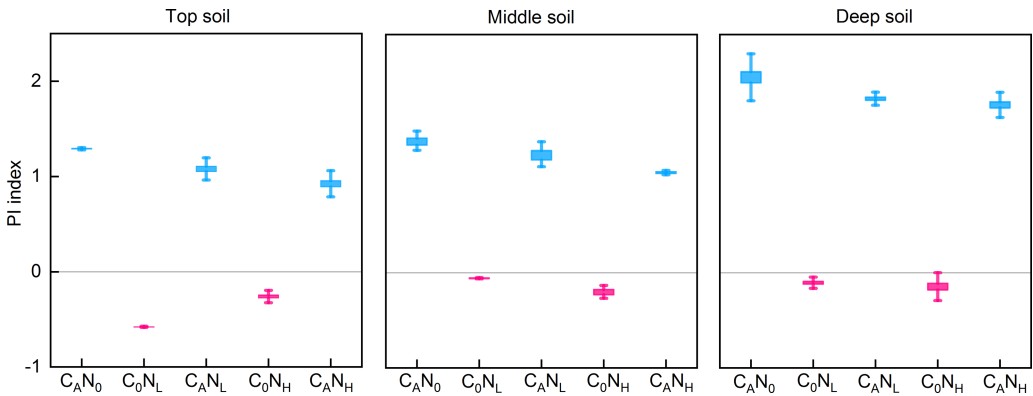

**Figure 3** **PI index from the top soil , middle soil, and deep soil in each treatment after 80days incuba-tion.** $C_0$, no-defoliation addition; $C_A$, defoliation addition; $N_0$, no-nitrogen input; $N_L$, low-nitrogen input; $N_H$, high-nitrogen input.

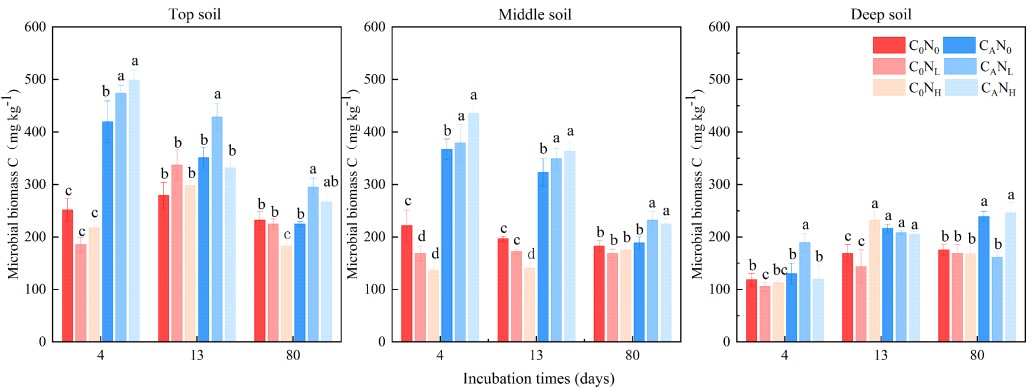

**Figure 4** **Soil microbial biomass C(MBC) at 4th, 13th, and 80th days after the start of the incubation in the top soil, middle soil, and deep soil.** $C_0$, no-defoliation addition; $C_A$, defoliation addition; $N_0$, no-nitrogen input; $N_L$, low-nitrogen input; $N_H$, high-nitrogen input. Different lowercase letters over the bar indicate significant difference at $p \leqq 0.05$ in the same incubation time.

depths. MBC in top and middle soils was higher than that in deep soils by 127.07% and 104.31%, respectively.

## Soil enzymes

The activity of soil catalase, alkaline phosphatase, and cellulase was enhanced, which was markedly changed by defoliation, nitrogen and soil depth, on the 4th day (Fig. 5 and Table 1). The activity of soil catalase was 32.95% and 36.94% higher, that of alkaline phosphatase was 84.64% and 83.03% higher, and that of cellulase was 174.67% and 251.12% higher under $C_A N_L$ and $C_A N_H$ treatment than under $C_0 N_0$. Nitrogen addition alone had different influences on activity of the tested soil enzymes in soils at the three depths. The activity of catalase, alkaline phosphatase, and cellulase was higher in top soils than in deep soils.

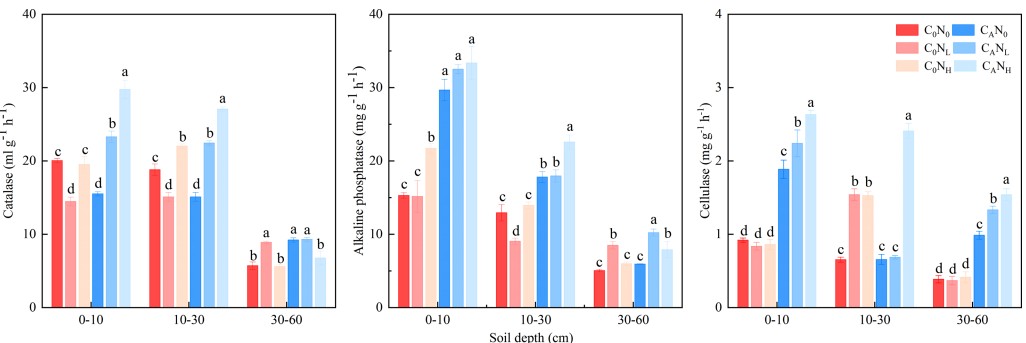

**Figure 5** **Soil enzyme activity on 4th day in the top soil, middle soil, and deep soil.** $C_0$, no-defoliation addition; $C_A$, defoliation addition; $N_0$, no-nitrogen input; $N_L$, low-nitrogen input; $N_H$, high-nitrogen input. Different lowercase letters over the bar indicate significant difference at $p \leq 0.05$ in the same soil depth.

## Soil bacterial diversity and community structure

During the first 7 days, $CO_2$ emissions rapidly surged after exogenous carbon addition. Mounting studies (*Tian et al., 2016a*; *Shen et al., 2021*; *Zhou et al., 2021*) reported that bacterial communities were changed from day 3 to day 15. Moreover, our study demonstrated that MBC peaked on the 4th day under defoliation and nitrogen addition. Accordingly, in order to evaluate the effect of defoliation, nitrogen, and soil depth on the soil microbial community structure in dryland cherry orchards, soil DNA was extracted on the 4th day of incubation to analyze the bacterial diversity. The results manifested that defoliation and nitrogen addition exerted no significant effects on the Chao1, Shannon, and Simpson index of soil bacteria (Fig. 6). However, the relative abundance of *Proteobacteria* markedly increased in soils at the three depths under defoliation and nitrogen addition, and that of *Acidobacteria* in deep soils was substantially reduced under both defoliation and nitrogen addition.

In our research, 48 phyla, 133 classes, 209 orders, 260 families, and 480 genera of bacteria were obtained through high-throughput sequencing of bacterial 16S rRNA. At the level of bacteria phyla (Fig. 7), the relative abundance of *Proteobacteria* and *Acidobacteria* was greater than 10.0% in communities, which were 37.84%–60.62% and 10.70%–24.53%, respectively. The relative abundance of *Proteobacteria* was 15.91% higher under $C_A$ treatment than under $C_0$ treatment and was 12.18% and 11.85% higher under $N_H$ and $N_L$ treatment than under $N_0$ treatment in deep soils. The relative abundance of *Acidobacteria* in deep soils was lower under $C_A N_H$ treatment than under $C_0 N_0$ treatment by 96.23%.

## DISCUSSION

### Effects of defoliation and nitrogen addition on $CO_2$ emissions in soils at different depths

As previously described (*Fontaine, Mariotti & Abbadie, 2003*), defoliation alone or both defoliation and nitrogen was added in our study to assess organsim matter decomposition and analyze the mechanism of "microbial mining". Our data unveiled that cumulative
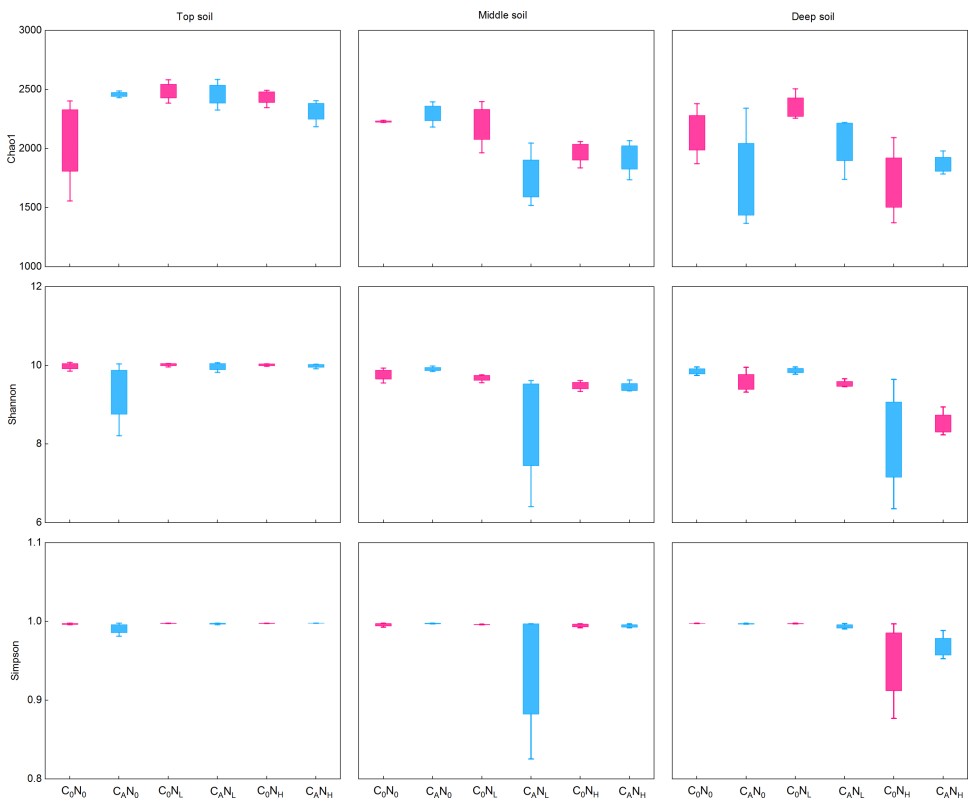

**Figure 6 Richness (Chao1) and diversity (Shannon and Simpson) of bacteria in the top, middle, deep soils on 4th day.** $C_0$, no-defoliation addition; $C_A$, defoliation addition; $N_0$, no-nitrogen input; $N_L$, low-nitrogen input; $N_H$, high-nitrogen input.

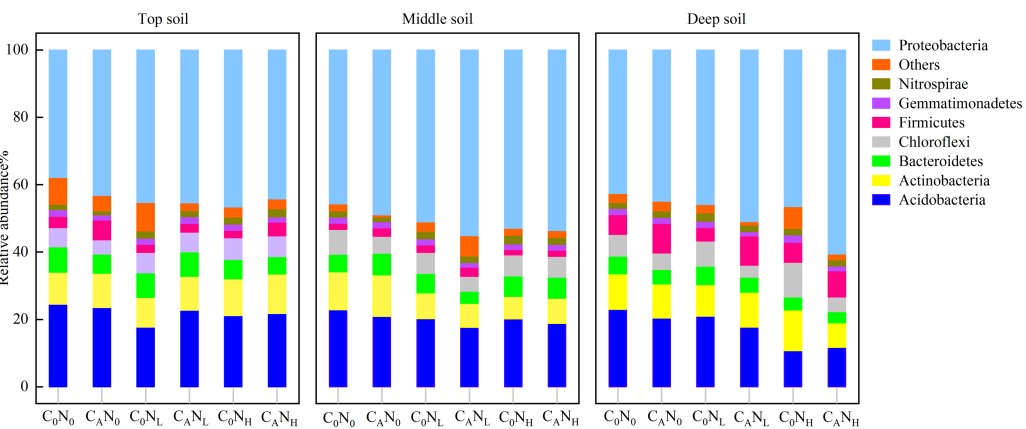

**Figure 7 Relative abundance of bacteria phyla in the top, middle, deep soil on 4th day.** $C_0$, no-defoliation addition; $C_A$, defoliation addition; $N_0$, no-nitrogen input; $N_L$, low-nitrogen input; $N_H$, high-nitrogen input.

$CO_2$ emissions under defoliation addition alone or combined with nitrogen addition were $1.94-2.78$ times higher than those under $C_0N_0$ treatment, which may be attributable to

the fact that defoliation and nitrogen addition increased the activity of soil microbes (Figs. 4 and 5), accelerated the conversion of microbial biomass (*Blagodatskaya & Kuzyakov, 2008*), and facilitated the secretion of extracellular enzymes to decompose soil organsim matter and then deciduous residues through microbes (*Allison et al., 2010*; *Burns et al., 2013*), thus resulting in $CO_2$ emissions in short-term incubation (*Cotrufo et al., 2015*; *Liu et al., 2017*; *Zhou et al., 2021*). These results are consistent with hypothesis (1). Additionally, our findings also elucidated that compared to defoliation addition alone, addition of both defoliation and nitrogen decreased $CO_2$ emissions and PI, which is ascribed to the "mining of nitrogen" (*Fontaine, Mariotti & Abbadie, 2003*). Moreover, nitrogen alone addition also reduced $CO_2$ emissions and caused negative PI, which is attributed to the "mining of nitrogen" (*Fontaine, Mariotti & Abbadie, 2003*). Our data also revealed that in soils with relatively low total nitrogen contents ($0.47-0.74$ g kg$^{-1}$), microbes were stimulated to decompose organsim matter to acquire the required nitrogen and induce $CO_2$ emissions under defoliation and nitrogen addition, similar to most research results (*Chen et al., 2014*; *Li et al., 2017*; *Fang et al., 2018a*; *Mason-Jones, Schmücker & Kuzyakov, 2018*; *Hicks et al., 2019*; *Liao, Tian & Liu, 2021*). However, our data showed no significant differences between low and high nitrogen, which indicated that the nitrogen demand of microbes could be saturated at low nitrogen levels. In addition, nitrogen addition alone reduced $CO_2$ emissions and then caused negative PI, which was concordant with the results of laboratory culture in the dark (*Wang et al., 2014*; *Fang et al., 2018a*) and outdoor testing (*Ginting et al., 2003*). Corresponding to the research with the DeNitrification-DeComposition (DNDC) model (*Chi & Chen, 2001*), *Grant et al. (2004)* also used the DNDC model to predict the impact of 50% $CO_2$ emissions on nitrogen. Therefore, nitrogen addition alone exerts little effect on soil respiration, whilst combination of nitrogen and organic carbon effectively promotes the decomposition of exogenous organic substances.

Soil depth is frequently associated with new increases in $CO_2$ emissions (*Meyer et al., 2018*; *Liao et al., 2020*). Throughout the incubation period, the $CO_2$ efflux rates and cumulative $CO_2$ emissions were higher in top and middle soils than in deep soils, which might be explained by the following factors: (i) high MBC and activities of soil catalase, alkaline phosphatase, and cellulase in top and middle soils (Figs. 4 and 5), (ii) relatively sufficient nutrients in top and middle soils, and (iii) suitable pH and favorable soil structure and properties in top and middle soils (*Wang et al., 2014*; *Tian et al., 2016b*), which provided advantageous conditions for the decomposition of deciduous residues by microbes. Conversely, PI was higher in deep soils than in top and middle soils in our study, concurrent with the result of a prior study (*Liao et al., 2020*). The proportion of residual organic carbon was calculated based on the input amount of organic carbon and emission amount of $CO_2$ in soils at different depths, which exhibited that the proportion of residual organic carbon was 88.88%, 88.05%, and 80.60% in top, middle, and deep soils, respectively. This result indicated that top and middle soils can retain a higher proportion of exogenous carbon than deep soils.

## Effects of defoliation and nitrogen addition on microbial activities and communities in different soils

Microbes have been extensively recognized to play a key role in soil carbon mineralization (*Herrmann & Bucksch, 2014*; *Chen et al., 2018*; *Li et al., 2018*). MBC and enzyme activities are usually considered measurable proxies for microbial decomposition (*Dorodnikov et al., 2009*; *Jiang et al., 2021*) and can be optimized with the shortest incubation time to minimize microbial growth and enzyme production during the measurement (*Shen et al., 2021*). Defoliation addition alone or combined with nitrogen addition elevated MBC, increased the activity of alkaline phosphatase, and cellulase, and caused positive PI in soils (Fig. 3). Defoliation shared significantly positive correlations with MBC and the activity of catalase, alkaline phosphatase, and cellulase in soils (Fig. 8). These data illustrated that microbes obtained available carbon and nitrogen from defoliation and nitrogen through various enzymes to meet their stoichiometric requirements (*Lashermes et al., 2016*). In our study, nitrogen increased microbial biomass and the activity of the tested enzymes in defoliation and nitrogen additions (Figs. 4 and 5) as a regulator in organic carbon mineralization, which was supported by many previous studies (*Frey et al., 2004*; *Cayuela, Sinicco & Mondini, 2009*; *Parajuli, Ye & Szogi, 2022*). We also observed that at the initial stage of culture, addition of both defoliation and nitrogen (especially both defoliation and high nitrogen) markedly enhanced MBC and enzyme activities but diminished $CO_2$ emissions when compared with defoliation addition alone, further confirming the "microbial mining" mechanism (*Fontaine, Mariotti & Abbadie, 2003*). Meanwhile, our results also unraveled that only nitrogen addition significantly decreased soil microbial biomass and changed the activity of the tested enzymes at the three depths, accompanied by reduced $CO_2$ emissions and negative PI. These findings illustrated that the effective combination of nitrogen and organic carbon could accelerate organism decomposition, improve soil quality and fertility, and provide guidance for production practice.

In our study, *Proteobacteria* and *Acidobacteria* were dominant bacteria in soils regardless of soil depth or exogenous inputs (Fig. 7). Defoliation and nitrogen addition did not significantly change soil bacterial diversity and abundance (Fig. 6), indicating the ecological properties of *Proteobacteria* and *Acidobacteria* in soils. Defoliation addition alone or combined with nitrogen addition markedly elevated the relative abundance of *Proteobacteria* and reduced the relative abundance of *Acidobacteria*, thereby causing high $CO_2$ emissions. This result suggested the association of *Proteobacteria* and *Acidobacteria* with $CO_2$ emissions. As a copiotrophic group, *Proteobacteria* grows rapidly, and its abundance is increased by relying on more labile carbon sources under nutrient inputs (*Ramirez, Craine & Fierer, 2012*). With increasing stimulation, more *Proteobacteria* can participate in the synthesis of exoenzymes to mineralize substrates (*Schimel, 2003*; *Fierer, Bradford & Jackson, 2007*), corresponding to the result that defoliation addition alone or combined with nitrogen addition substantially augmented MBC and enzyme activities, which was consistent with hypothesis (2). Our data revealed that $CO_2$ emissions were significantly positively correlated with *Proteobacteria* (Fig. 8). In addition, *Nottingham et al. (2009)* observed that microbial biomass increased in the first step of plant residue decomposition, which was caused by labile C rather than macromolecular compounds.

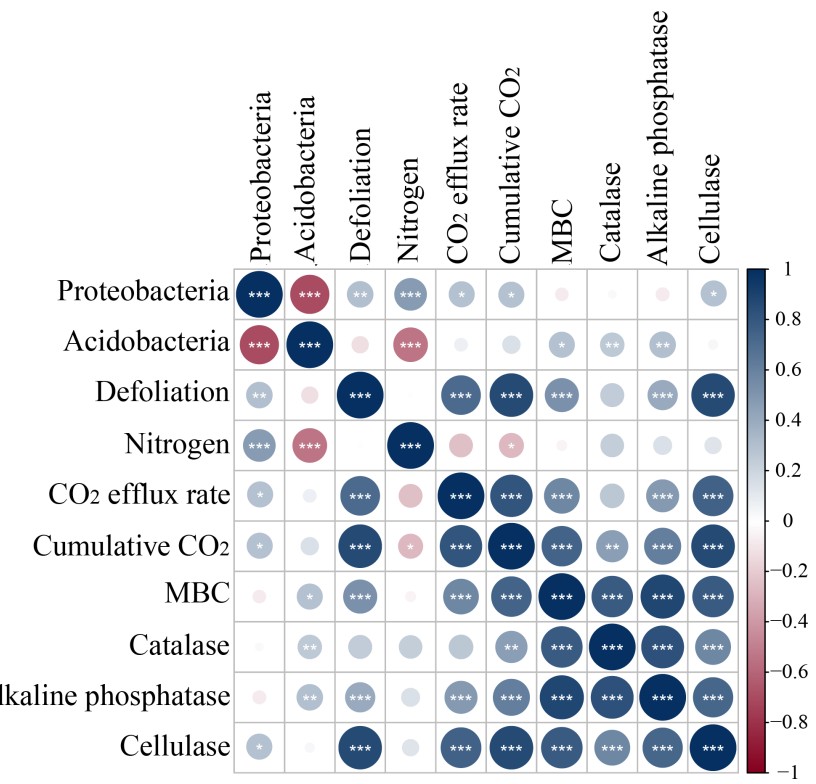

**Figure 8** **Spearman correlation relationship among the relative abundance of bacteria symbols coded on 4th day.** Red negative; Blue positive. Darker colors indicate stronger correlationships, asterisks indicate significant effects (***$p < 0.001$, **$p < 0.01$, *$p < 0.05$).

Altogether, these data suggest *Proteobacteria* as a main participant in organic carbon mineralization in cherry orchards. However, *Acidobacteria* (an oligotrophic group) could be colonized on mineral surfaces under rich nutrient conditions (*Nemergut et al., 2010*; *Ramirez, Craine & Fierer, 2012*).

Furthermore, our PLS-DA analysis (Fig. 9) demonstrated that bacterial communities were clearly separated in top and deep soils, which can be explained by the fact that nutrient composition determines the distribution of oligotrophic and hypertrophic bacteria (*Fierer, Bradford & Jackson, 2007*; *Whitman et al., 2016*), consistent with hypothesis (3). Compared to top soils, deep soils had the highest relative abundance of *Proteobacteria* under both defoliation and nitrogen addition, which can be attributed to the highest PI in deep soils that is related to different microbial products (*Kögel-Knabner, 2017*), soil pH (*Madsen & Munk, 1987*; *Silveira et al., 2008*; *Takele, Chimdi & Abebaw, 2014*), and soil nutrient (*Madsen & Munk, 1987*). Therefore, soil microbial communities may be partially determined by soil properties. Furthermore, fungi have also been reported to directly participate in organic carbon mineralization. Accordingly, further research is warranted to determine how fungal communities respond to defoliation and nitrogen added in soils of cherry orchards in northwest China.

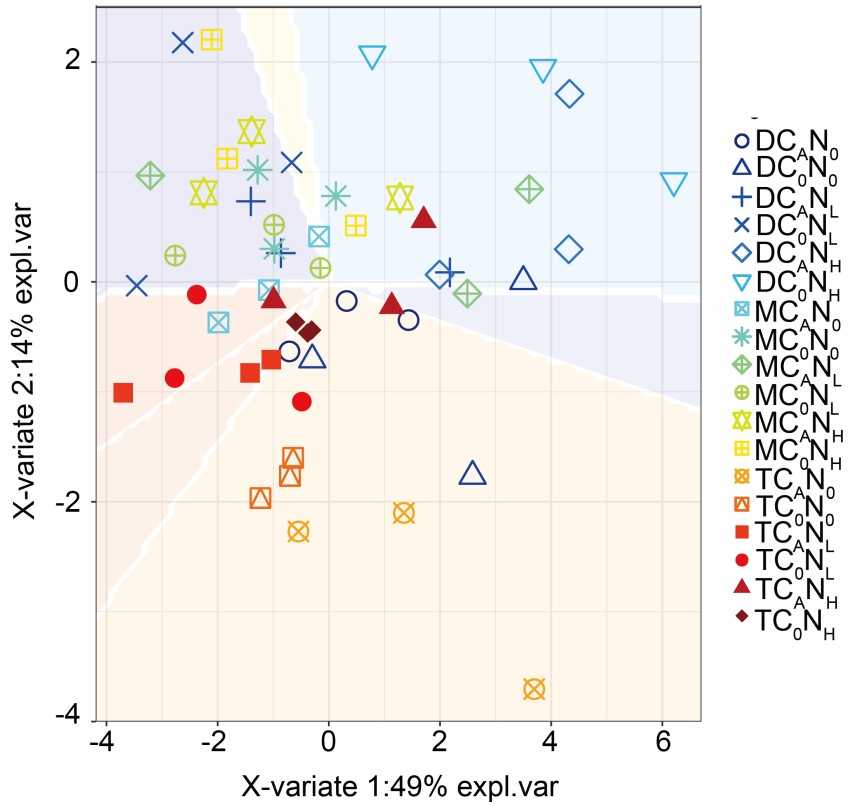

**Figure 9** **PLS$_{DA}$ of the bacterial communities on 4th day.** T, top soil; M, middle soil; D, deep soil; $C_0$, no-defoliation addition; $C_A$, defoliation addition; $N_0$, no-nitrogen input; $N_L$, low-nitrogen input; $N_H$, high-nitrogen input.

## CONCLUSION

In the early stages of the incubation period, defoliation and nitrogen addition increased MBC, the activity of catalase, alkaline phosphatase, and cellulase, and $CO_2$ emissions and resulted in positive PI in soils at the three depths of dryland cherry orchards. Nitrogen addition alone contributed to a significant reduction in MBC and $CO_2$ emissions and negative PI. Meanwhile, defoliation and nitrogen addition markedly elevated the relative abundance of *Proteobacteria* and reduced the relative abundance of *Acidobacteria* in soils at the three depths, particularly deep soils. Moreover, *Proteobacteria* was significantly correlated with defoliation, nitrogen, $CO_2$ emissions, and soil cellulase enzyme activities. In conclusion, this study unraveled the separate and interactive effects of defoliation and nitrogen addition on $CO_2$ emissions, soil microbial activities, and soil microbial communities composition at the soil depth scale. Likewise, this study demonstrated that both defoliation and nitrogen addition simulated exogenous carbon decomposition, affected soil microbial communities and improved soil microbial activities, ultimately enhancing soil quality in dryland cherry orchards.

### Funding

This research was supported by the Key Program of the Double First-Class Scientific Researches in Gansu (GSSYLXM-08), the National Science Funds of Gansu Province (grant no. 21JR11RE030), and a school project grant from Tianshui Normal University (grant no. CXJ2021-03). The funders had no role in study design, data collection and analysis, decision to publish, or preparation of the manuscript.

### Grant Disclosures

The following grant information was disclosed by the authors:
Key Program of the Double First-Class Scientific Researches in Gansu: GSSYLXM-08.
National Science Funds of Gansu Province: 21JR11RE030.
Tianshui Normal University: CXJ2021-03.

### Competing Interests

The authors declare there are no competing interests.

### Author Contributions

- Jing Wang conceived and designed the experiments, performed the experiments, analyzed the data, authored or reviewed drafts of the article, and approved the final draft.
- Yibo Wang conceived and designed the experiments, performed the experiments, analyzed the data, authored or reviewed drafts of the article, and approved the final draft.
- Ruifang Xue conceived and designed the experiments, analyzed the data, prepared figures and/or tables, and approved the final draft.
- Dandan Wang conceived and designed the experiments, analyzed the data, prepared figures and/or tables, and approved the final draft.
- Wenhui Nan conceived and designed the experiments, prepared figures and/or tables, and approved the final draft.

### Data Availability

The data is available at NCBI SRA: PRJNA896137.

### Supplemental Information

Supplemental information for this article can be found online at http://dx.doi.org/10.7717/peerj.15276#supplemental-information.

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
