# Peer review of "Effects of defoliation and nitrogen on carbon dioxide (CO2) emissions and microbial communities in soils of cherry tree orchards"

_PeerJ, doi:10.7717/peerj.15276_

## Round 0.1 · original submission · Major Revisions

· Academic Editor

Major Revisions

Dear Dr Wang

In light of the reviewers' comments, you have to extensively revise the manuscript with a focus on English language, and the specific comments raised by the referees. You are expected to submit the revised draft within the stipulated time. The reviewers' comments are appended below:

Reviewer 1 ·

Basic reporting

study is comprehensive but poor presentation

Experimental design

Experimental design is very good to see the impact of defoliation and nutrient (N) supply.

Validity of the findings

Manuscript Title: “Effects of exogenous carbon and nitrogen on priming effect and microbial communities on rain-fed cherry orchards soils”
1. More information can be added to the abstract and introduction part related soil depth, microbial diversity and soil health indicators
2. Limited content in introduction part and need to rewrite with recent references.
3. Introductory part is partially connected with study performed please improve it?
4. Lots of errors in spelling, spacing, sentences and grammar are needs to be corrected /rephrased?
5. Author should focus on to analyze and interpret datasets in a meaningful way integrating soil health, genomic, bioinformatics and statistics
6. If possible , soil enzymatic assay for dehydrogenase, FDA & acid & alkaline phosphatase activity should be done for different soil depth
7. The description of your work is very detailed at some points; however, there are sentences that are not clearly stated and it is hard to know the meaning and the input towards the main goal.
8. Pay attention to the quality of figures, grammar, spelling of words and unclear sentences throughout the manuscript
9. Why proteobacteria abundance is very significant for priming effect please justify?
10. Role of pH at different soil strata should be mentioned?
11. The research article comprehensively explained in results but poorly justified. Limited discussion part is the main drawback of this article.
Recommendation: Accepted with major corrections

Additional comments

Nil

Reviewer 2 ·

Basic reporting

Research paper ‘Effects of exogenous carbon and nitrogen on priming effect and microbial communities on rain-fed cherry orchards soils’ by Wang et al., is based on the effect of defoliation of cherry tree leaves on the CO2 emission and microbial communities in an incubation study.
The idea/ research theme is quite interesting but the title is not justifying the content of the manuscript. Title should reflect the content so author may rewrite as- ‘Effect of defoliation on the CO2 emission and microbial communities of soil under cherry tree orchard’. ‘Effect of…..on priming effect…’ is not justifying its meaning.
Authors can present this paper in more effective way. There are lots of problems in English language some of them I am mentioning but authors need to relook whole manuscript for this. Here I am mentioning some points need to be addressed
Abstract line 14 factors facters
17- should indicate depth in some units not just top- middle….
24, 34, 37, 43 spacing
25 PI ??? at the first appearance and in the abstract full form of all abbreviation should be mentioned.
27- in all treatments not each
Grammar- 38 were are, It was- It is, 42- not was , is 43-44- rewrite this sentence , 49 –soils of different depth … collected, 61 –of in
83 –no-defoliation addition should no-defoliation or without defoliation
96 –jars was
101 why the DNA was extracted on after 4 days incubation? Is any specific reason behind this ?
108 system??
169 cincubation
195 –miaddle
208- bacteriophyta this is not generally used term
211- recheck is it significantly different??
219 by …%
262- Fontain or Fontaine
267 Fieter or Fierer
277 I could not found positive correlation in this figure for Proteobacteria and CO2 efflux
278-279- how author can found this relation in fig 7 b.
figure 7 is not clear.
Some detail of methods should be added in abstract.
References are region specific.
Author can calculate how much organic material and C (also N) added via defoliation and how much emitted and how much left in soil. Why there is difference in mineralization with depths should more elaborate.

Experimental design

No comment

Validity of the findings

No comment

Additional comments

Manuscript should be revised extensively for content and language both.

---

## Round 0.2 · Minor Revisions

· Academic Editor

Minor Revisions

Dear Dr. Wang

Thank you for your submission to PeerJ.

It is my opinion as the Academic Editor for your article - Effects of defoliation and nitrogen on the CO2 emission and microbial community of soil under cherry tree orchards - that it requires a number of Minor Revisions which mainly relate to English language and the way sentences are written. I am specifically pointing out the following changes to the manuscript to further improve its quality:

1. There still exists good scope for improving the English language and the way sentences are written.

2. The words are joined together at several places throughout the manuscript.

3. At several places in the running sentences, the letters should be in small case while they are in upper case.

4. Several words throughout the text are wrongly spelled, e.g., orcads in line 34, and alterd in line 81.

5. There are many sentences that need to be revised for a better meaning; for instance but not limited to the sentences in lines 17-19, lines 20-22, lines 27-29, lines 164-167, lines 168-169, lines 209-210 and lines 244-245 among others.

In light of these discrepancies, you are advised to carefully revise the manuscript to enhance its readability.

Reviewer 1 ·

Basic reporting

Although authors revised the manuscript but still authors needs to pay attention in recent findings in introduction section of the manuscript.

Experimental design

As per the suggestions authors have incorporated in the desired information like soil enzymatic assay, sampling of different depth of soil samples etc. Author showed the focus on to analyze and interpret datasets in a meaningful way for integrating soil health, genomic, bioinformatics and statistics.

Validity of the findings

Authors can also improve the discussion of the manuscript as per findings that are not sufficient in present form. Recent finding of the effects of carbon and nitrogen and their combined impact in environment can be discussed well for more improvement of the manuscript. Although, authors have explained the results with proper justifications.

---

## Round 0.3 · accepted · Accept

· Academic Editor

Accept

Dear Dr. Wang

Based on the perusal of the revised manuscript, your paper entitled- Effects of defoliation and nitrogen on the CO2 emissions and microbial communities of soil under cherry tree orchards - has been Accepted for publication.